# Shared Molecular Targets in Parkinson’s Disease and Arterial Hypertension: A Systematic Review

**DOI:** 10.3390/biomedicines10030653

**Published:** 2022-03-11

**Authors:** Delia Tulbă, Mioara Avasilichioaei, Natalia Dima, Laura Crăciun, Paul Bălănescu, Adrian Buzea, Cristian Băicuș, Bogdan Ovidiu Popescu

**Affiliations:** 1Department of Neurology, Colentina Clinical Hospital, 020125 Bucharest, Romania; delia.tulba@umfcd.ro (D.T.); mioara.avasilichioae@stud.umfcd.ro (M.A.); natalia-ioana.dima@rez.umfcd.ro (N.D.); laura.c.craciun@gmail.com (L.C.); 2Department of Clinical Neurosciences, “Carol Davila” University of Medicine and Pharmacy, 020021 Bucharest, Romania; 3Colentina–Research and Development Center, Colentina Clinical Hospital, 020125 Bucharest, Romania; paul.balanescu@umfcd.ro (P.B.); cristian.baicus@umfcd.ro (C.B.); 4Clinical Research Unit, RECIF (Réseau d’Epidémiologie Clinique International Francophone), 020125 Bucharest, Romania; 5Department of Internal Medicine, “Carol Davila” University of Medicine and Pharmacy, 020021 Bucharest, Romania; catalin.buzea@umfcd.ro; 6Department of Cardiology, Colentina Clinical Hospital, 020125 Bucharest, Romania; 7Department of Internal Medicine, Colentina Clinical Hospital, 020125 Bucharest, Romania; 8Laboratory of Cell Biology, Neurosciences and Experimental Myology, “Victor Babeș” National Institute of Pathology, 050096 Bucharest, Romania

**Keywords:** Parkinson’s disease, arterial hypertension, molecular targets

## Abstract

(1) Background: Parkinson’s disease and arterial hypertension are likely to coexist in the elderly, with possible bidirectional interactions. We aimed to assess the role of antihypertensive agents in PD emergence and/or progression. (2) We performed a systematic search on the PubMed database. Studies enrolling patients with Parkinson’s disease who underwent treatment with drugs pertaining to one of the major antihypertensive drug classes (β-blockers, diuretics, angiotensin-converting enzyme inhibitors, angiotensin receptor blockers and calcium-channel blockers) prior to or after the diagnosis of parkinsonism were scrutinized. We divided the outcome into two categories: neuroprotective and disease-modifying effect. (3) We included 20 studies in the qualitative synthesis, out of which the majority were observational studies, with only one randomized controlled trial. There are conflicting results regarding the effect of antihypertensive drugs on Parkinson’s disease pathogenesis, mainly because of heterogeneous protocols and population. (4) Conclusions: There is low quality evidence that antihypertensive agents might be potential therapeutic targets in Parkinson’s disease, but this hypothesis needs further testing.

## 1. Introduction

### 1.1. Background/Rationale

Parkinson’s disease (PD) is the most common neurodegenerative movement disorder, with an incidence ranging from 5 to 35 cases per 100,000 population yearly and a prevalence that increases from 1% of people aged 45–54 to 4% of men older than 85 years [1]. Over recent years, the biomedical and economic burden of PD has increased dramatically as a result of population ageing [2]. In his “Essay on the Shaking Palsy”, James Parkinson made some “considerations respecting the means of cure” and concluded that “nothing direct and satisfactory has been obtained.” [3]. More than 200 years later, this statement is still true: despite extensive research efforts, there are no curative/disease-modifying therapies or neuroprotective interventions in PD. 

Since the prevalence of both PD and arterial hypertension increases with age, they are likely to coexist in the elderly, with possible bidirectional interactions at various levels. The interference between antihypertensive agents and PD emergence and/or progression has recently become a subject of great interest and debate among researchers and health care professionals. The hypothesis that antihypertensive drugs might have neuroprotective or disease-modifying properties in PD has been tested in animal models and clinical trials, but no final verdict has been reached. Intriguingly, all the molecular targets of antihypertensive medication have been proposed to interfere with the process of neurodegeneration in PD, but a unifying and consensual view is still lacking. Considering the availability and wide use of antihypertensive therapies, it is of utmost importance to determine whether and to what extent they modulate the pathogenesis and progression of PD. 

#### 1.1.1. Beta-Blockers (BBs)

The notion that beta-blocking agents might be helpful in the treatment of PD was suggested a long time ago in a case report of a patient undergoing deep brain stimulation surgery whose rigidity was transiently reduced after an intravenous dose of metoprolol (via the temporary suppression of bursting spiking activity in the subthalamic nucleus) [4]. On the other hand, there have been reports of an increased risk of PD among propranolol users, but the evidence for a causal link between the two is rather tenuous [5]. 

While their full workings are yet to be clarified, β_1_ and β_2_ adrenoreceptors are expressed in the brain—albeit unevenly among cell types: β_1_ receptors are mostly expressed in astrocytes, microglia (tentatively) and possibly neurons and endothelial cells, whereas β_2_ receptors are probably expressed in neurons and astrocytes but definitely expressed in other cell types, at least in murine models [6]. Regionally speaking, β_2_ adrenoreceptors are reportedly expressed in the substantia nigra and cortex [7], with possible implications for PD emergence and/or progression [6,7,8]. 

#### 1.1.2. Diuretics

Although not so extensively investigated, diuretics supposedly interfere with neurodegeneration. In cellular models of Alzheimer’s disease, indapamide and hydrochlorothiazide were able to suppress the production of amyloid-β peptide and improve its clearance [9,10]. Moreover, it has been suggested that higher serum potassium levels resulting from exposure to potassium-sparing diuretics might mitigate cognitive decline [11]. It is no wonder that these assumptions sparked excitement regarding a possible cytoprotective role of diuretics in other proteinopathies such as PD. The mechanisms involved are quite heterogeneous and not fully elucidated [9,10,12], but the hypothesis of inhibition of α-synuclein aggregation is worth mentioning [13]. 

#### 1.1.3. Calcium-Channel Blockers (CCBs)

Voltage-gated calcium channels (VGCCs) act as sensors that convert extracellular signals into intracellular action. As such, calcium ion flows modulate neurotransmitter release, muscle contraction, hormone secretion and gene expression. VGCCs consist of high voltage-activation (HVA) types—including the L-type (Ca_v_1 family), R-, P-/Q-, and N-type (Ca_v_2 family—and low voltage-activation (LVA) types, namely the T-type (Ca_v_3 family). Ca_v_1.2 is the most common isoform expressed in the brain, heart, smooth muscle, and pancreas, whereas Ca_v_1.3 is rather confined to the neuronal system [14], being involved in the pathogenesis of PD [14,15,16,17,18,19]. Yet, other genome-based studies appear to indicate that stimulating L-type calcium channels might help boost dopamine synthesis [20].

The relevant literature is rife, with reports of antihypertensive CCBs being effective in neurodegenerative diseases. Nimodipine, for instance, a poor blood-brain barrier (BBB) crossing drug, has been magnetically nano-delivered to PD rats, to good effect [21]. Felodipine, to give another example, an antihypertensive medication operating as an L-type channel blocker with BBB penetrating properties, has recently been shown to act as an autophagy-inducer and α-synuclein cleaner in mice at plasma doses like those seen in patients taking the drug for its antihypertensive potence [22]. Conversely, L-type CCBs such as nifedipine and amlodipine, which can cross the BBB [23], have been found to trigger manifestations in the spectrum of the de Melo-Souza’s parkinsonian syndrome that are classically associated with the T-type calcium channel blockers—cinnarizine and flunarizine [24]. Nevertheless, the mechanisms underlying the effect of CCBs on PD pathogenesis are far from being elucidated [25,26,27,28,29]. 

#### 1.1.4. Angiotensin-Converting Enzyme Inhibitors (ACEIs) and Angiotensin Receptor Blockers (ARBs)

Renin-angiotensin-aldosterone system (RAAS) is an endocrine signaling system that regulates blood pressure and water and electrolyte balance. Its action is surveyed by a complex neural pathway (central RAAS) that also engages angiotensin II [30,31]. Since peripheral angiotensin II does not cross the BBB, it accesses the circumventricular organs (which lack a BBB) such as the subfornical organ and vascular organ of the lamina terminalis, where it provides signals related to low blood volume [31], modulating the activity of brain RAAS. Neurons in these areas further project to and release angiotensin II to various hypothalamic nuclei (e.g., median preoptic area and paraventricular nucleus), eliciting drinking behavior [31]. Apart from its role as a hormone, this pathway unravels the capacity of angiotensin II to act as a neurotransmitter. 

Interestingly, all the mediators and effectors of the RAAS are widely distributed in the central nervous system inside the BBB (both by already active promoter regions of their genes and de novo synthesis in the brain)—angiotensinogen is produced within astrocytes (where its encoding mRNA is mainly found) where it secretes various neuroactive peptides, renin is expressed in both neurons and astrocytes, angiotensin converting enzyme is mainly localized in the endothelium of cerebral vasculature, choroid plexus, area postrema and other circumventricular organs as well as nigrostriatal pathway and basal ganglia, whereas angiotensin type II receptors (AT1 and AT2, the former abundant in the brain) that bind angiotensin type II can be found in the neurons, astrocytes, oligodendrocytes, and microglia of different brain areas [32,33,34]. Notably, angiotensin II is secreted in the brain independently from the peripheral sources [35]. Regulatory interactions between the central RAAS and dopaminergic system have been described in the substantia nigra and striatum [36]. In animal models, dopamine depletion exerts compensatory activation of central RAAS [36], whereas enhanced levels of angiotensin II supposedly play a synergistic role in the pathogenesis and progression of PD [37]. Nevertheless, the role of ACEIs and ARBs in this matter is still unknown [32,34,35,36,37]. 

### 1.2. Objectives

We aimed to assess the role of antihypertensive drugs in PD pathogenesis (both emergence and progression). We hypothesized that they might exert neuroprotective and/or disease-modifying effects in a dose and time-dependent manner. 

## 2. Materials and Methods

### 2.1. Protocol

The protocol for this systematic review was conceived a priori based on the PRISMA 2020 Statement which comprises a checklist of 27 items. It was submitted for registration in the PROSPERO International prospective register of systematic reviews (ID: 314157, status: waiting for approval). We designed a systematic review centered on the following research question:

“Does hypertension medication pertaining to major antihypertensive drug classes exert neuroprotective and/or disease-modifying effects in a dose and time-dependent manner in adult patients with sporadic PD regardless of blood pressure values?”

The defined target population (P) consisted of adult patients diagnosed with sporadic PD. The intervention (I) was intake of drugs pertaining to one of the major antihypertensive drug classes (i.e., BBs, diuretics, CCBs, ACEIs, and ARBs,), in any regimen. The comparator (C) involved subjects not receiving the intervention. The outcome (O) was either the occurrence (in patients receiving antihypertensive drugs before the motor onset and diagnosis of PD) or the progression of PD (in patients receiving antihypertensive drugs afterwards). 

We performed a systematic search on the PubMed database on 6 June 2021, searching for studies that enrolled patients with PD who underwent treatment with antihypertensive drugs (prior to or after the diagnosis of PD). We conducted the search again on 10 July and 28 December 2021 in order to identify articles published meanwhile. Articles were included from inception. We applied one filter (Humans) and used the following search strategy:

(“Vasodilator Agents” [MeSH Terms] OR (“Angiotensin-Converting Enzyme Inhibitors” [MeSH Terms] OR “Angiotensin II Type 1 Receptor Blockers” [MeSH Terms] OR “Angiotensin Receptor Antagonists” [MeSH Terms] OR “Calcium Channel Blockers” [MeSH Terms] OR “Dihydropyridines” [MeSH Terms] OR “Diuretics” [MeSH Terms] OR “Adrenergic beta-Antagonists” [MeSH Terms] OR “Sympatholytics” [MeSH Terms])) AND (“parkinson*” [Title/Abstract] OR “Parkinson Disease” [MeSH Terms] OR “Parkinsonian Disorders” [MeSH Terms])

Studies that were cited in review articles or clinical trials identified through this search were also considered for inclusion. 

Inclusion criteria were: any clinical study (either observational or interventional, prospective or retrospective) enrolling adult patients diagnosed with sporadic PD (either fulfilling the UK Parkinson’s Disease Society Brain Bank Diagnostic Criteria [38] or the Movement Disorder Society Clinical Diagnostic Criteria for Parkinson’s disease [39]) who took medication (before or after PD diagnosis) pertaining to one of the major antihypertensive drug classes, as defined by the “ESC/ESH Guidelines for the management of arterial hypertension” [40], namely BBs, diuretics (thiazides and thiazide-like diuretics), CCBs (dihydropyridines), ACEIs and ARBs, and which assessed their potential neuroprotective and/or disease modifying effect in PD. These five major drug classes are recommended for the treatment of arterial hypertension based on “proven ability to reduce blood pressure, evidence from placebo-controlled studies that they reduce cardiovascular events, and evidence of broad equivalence on overall cardiovascular morbidity and mortality, with the conclusion that benefit from their use predominantly derives from blood pressure lowering” [40]. History of arterial hypertension was not mandatory for inclusion. Exclusion criteria were: unavailable data on PD emergence or progression, other types of parkinsonism (e.g., drug-induced parkinsonism, atypical parkinsonism, vascular parkinsonism), other classes of antihypertensive drugs (i.e., non-dihydropyridines, alpha-blockers, centrally active drugs, vasodilators, loop and potassium-sparing diuretics), drugs with similar mechanisms not used in the treatment of arterial hypertension (e.g., cinnarizine as a CCB), unavailability of full-text, other languages than English, surveys or self-reported effects.

### 2.2. Study Appraisal 

Four authors evaluated the studies (L.C. independently, D.T., M.A., and N.D. together) and included the eligible ones. They covered the abstracts based on inclusion criteria; the ones eligible were subsequently assessed full text. Differences were discussed with other authors until reaching a consensus. Duplicates were removed. 

Data collection was performed manually by four authors (L.C., M.A., N.D., and D.T.). Information regarding study population (number of PD patients, sex, mean age), PD progression and staging (PD duration, Hoehn and Yahr stage, Unified Parkinson’s Disease Rating Scale (UPDRS) score, antiparkinsonian drugs—types and dosage), control group (number of subjects), arterial hypertension (yes/no, grading, risk stratification, duration, hypertension-mediated organ damage), antihypertensive drugs (regimen: duration, dosage, administration) were extracted and introduced in the database. 

We divided the outcome into two categories, depending on the timeframe of antihypertensive drug intake relative to PD onset (before or afterwards)—neuroprotective (i.e., the antihypertensive drug is a protective factor for PD occurrence) or disease-modifying (i.e., the antihypertensive drug has a beneficial effect on the course of PD) effect. Two confounders were carefully assessed: drug-induced parkinsonism and symptomatic effect. Drug-induced parkinsonism is classically described as a symmetric parkinsonism without tremor at rest [41], as opposed to idiopathic PD in which bilateral symmetric parkinsonism is a red flag [39]. However, since clinical manifestations alone cannot always reliably differentiate drug-induced parkinsonism from idiopathic PD, clinical history is of utmost importance. “Treatment with a dopamine receptor blocker or a dopamine-depleting agent in a dose and time-course consistent with drug-induced parkinsonism” is an absolute exclusion criterion for PD [39]. Apart from the well-known neuroleptics and antiemetics, there are other drugs that deplete dopamine or elicit dopamine antagonist activity, some of them pertaining to the antihypertensive class (e.g., α-methyldopa, verapamil, diltiazem, captopril) [42]. These rare cases of drug-induced parkinsonism should not be confused with idiopathic PD following exposure to antihypertensives (i.e., antihypertensive agent as a risk factor for PD), so we tried to differentiate them considering our exclusion criteria. Another issue concerns the disease-modifying effect (which means a lasting effect) which should not be confused with a symptomatic effect (which means a transient and reversible effect on PD symptoms). Usually the disease-modifying effect is quantified with the change in UPDRS score from baseline to a certain period (preferably, longer periods of time). However, since UPDRS scale also reflects the symptomatic effect of antiparkinsonian drugs, the scale should be performed in an “OFF” state to assess the real disability of the patients. 

## 3. Results

The search on PubMed database identified 456 results, with 0 duplicates removed. Twenty-four additional studies were identified in review articles (both narrative and synthesis reviews) and clinical studies and were subsequently included. After screening by title and abstract, 459 studies were excluded. Those accepted were consequently read full-text and 20 studies fulfilled the inclusion criteria, as illustrated in the flow chart (Figure 1). All the studies included were analytical, both observational (case-control and cohort studies, either retrospective or prospective) and interventional (randomized controlled trials). 

Fifteen studies investigated whether the exposure to antihypertensive agents is a protective factor for PD emergence (neuroprotective effect—Table 1), whereas five studies explored the effect of antihypertensive therapy on PD progression (disease-modifying effect—Table 2). 

### 3.1. BBs

Ten studies addressed the effect of BBs on PD emergence, whereas one study investigated their effect on PD progression.

#### 3.1.1. BBs as Neuroprotective Treatment

In a retrospective case-control study, Hopfner et al. evaluated 2790 PD patients and 11,160 matched controls and identified an increased risk of PD among subjects with previous use of β_2_-antagonists (cases = 407, controls = 1.488); the association was stronger for short-term use (<1 year; OR = 1.97; 95% CI, 1.70–2.28) compared to long-term use (≥3 years; OR = 1.28; 95% CI, 1.10–1.47), suggesting a reverse causation (meaning that early stage PD symptoms such as tremor trigger the prescription of BBs, rather than BBs eliciting PD) [43]. 

Koren et al. showed that the risk of PD was increased among patients exposed to BBs during the years prior to PD diagnosis (adjusted HR = 1.51; 95% CI, 1.28–1.77), in a time and dose-dependent fashion, even after adjusting for other factors known to increase (gender, age) or decrease (cigarette smoking, cholesterol, statin use) the risk of PD [44]. In this prospective cohort study, they evaluated 145,098 patients who received BBs and 1,187,151 who did not [44]. Moreover, they excluded patients with BBs prescribed for benign tremor [44]. 

In a large, population-based case-control study (48,295 incident PD cases, 52,324 controls), Nielsen et al. found the use of propranolol to be associated with greater PD risk (OR = 3.62; 95% CI, 3.31–3.96) [45]. However, the risk dropped prominently after adjusting for tremor prior to PD diagnosis/reference and with greater lagging of propranolol exposure [45]. When simultaneously adjusting for tremor and applying the maximum lag of 18 months, the OR dropped by 70% and was close to null [45]. Conversely, metoprolol and carvedilol were associated with lower PD risk (OR = 0.94; 95% CI, 0.91–0.97 and OR = 0.77; 95% CI, 0.73–0.81, respectively) [45]. 

Gronich et al. conducted a nested case-control study in a cohort of 1,762,164 subjects without a diagnosis of PD and followed them for 13 years [46]. During follow-up, 11,314 patients developed PD and were matched with 113,140 controls. In contrast to selective β_1_-antagonists, the use of non-selective β-antagonists was associated with an increased risk of PD (RR = 1.00; 95% CI, 0.95–1.05 and RR = 2.04; 95% CI, 1.90–2.20, respectively) [46]. The positive association of PD emergence with nonselective BBs use was consistent and robust in different sensitivity analyses [46]. 

In a retrospective case-control study, among 3637 subjects with PD, Becker et al. identified 1168 (32.1%) who had used a BB prior to the index date and found no association between the use of BBs and risk of PD (adjusted OR = 1.16; 95% CI, 0.95–1.41) [47]. However, current short-term use of BBs (in users of <10 BBs prescriptions, without recorded cardiovascular disorders) was associated with an increased risk of PD emergence, also likely due to a confounding by indication (patients with early symptoms of PD such as tremor receiving symptomatic treatment with BBs) [47]. 

In a case-control study including 9127 patients with PD and a matched number of subjects without PD, Warda et al. inquired the incidence of PD as a function of the use of antihypertensive drugs [48]. They designed three models—model I: once used vs. never used, model II: at least three years of therapy vs. less than three years of therapy and model III: effect per therapy year—and found no association between BBs use and PD incidence after adjusting for co-diagnoses such as diabetes mellitus, coronary disease, hypertension, arrhythmias, heart failure, renal failure, stroke, and depression (model I: OR = 1.04; 95% CI, 0.97–1.11; model II: OR = 0.93; 95% CI, 0.87–1.00; model III: OR = 1.00; 95% CI, 1.00–1.01) [48]. 

Germay et al. also investigated the relationship between β-adrenoreceptor drug exposure (both agonists and antagonists) and PD occurrence in a nested case-control study [49]. Among the 2225 incident PD patients, no significant association was found between PD and BBs, except from propranolol (adjusted OR = 2.11; 95% CI, 1.38–3.23) [49]. Since the association became nonsignificant with a cumulative use time ≥6 months, a potential protopathic bias was proposed [49]. 

Although mainly focusing on the effect of CCBs on PD emergence (see below), Ritz et al. also found that the use of BBs was associated with a slightly increased risk of PD when employing a 2-year lag (OR = 1.29; 95% CI, 1.13–1.48); however, in the 5-year lagged analysis, the association diminished considerably (OR = 1.16; 95% CI, 0.99–1.37), which suggests a reverse causation [50].

Ton et al. investigated the risk of PD associated with BBs in a population-based case control study of 206 patients with a new diagnosis of PD and 383 controls without neurodegenerative disorders [51]. They found no association with PD risk among BBs users in terms of duration, dose, number of prescriptions, or pattern of use (adjusted OR = 1.20; 95% CI, 0.71–2.03) [51].

Giorgianni et al. performed a large cohort study with a nested case-control approach, assembling a cohort of 230,884 patients; during follow-up, 8604 patients developed PD and were matched to 86,040 controls [52]. An increase of 45% in the risk of PD with ever use of BBs compared to non-use was noticed (RR = 1.45; 95% CI, 1.37–1.54) [52]. However, the rate was greatest with less than one year of cumulative duration of use and decreased with longer exposure, showing no risk of PD with more than five years of use of BBs [52].

#### 3.1.2. BBs as Disease-Modifying Treatment

In a single-center, retrospective cross-sectional study, Laudisio et al. found that BBs did not lower the number of falls occurring in the last 12 months in patients with PD (*p* = 0.291) [53]. 

### 3.2. Diuretics (Thiazides and Thiazide-like Diuretics)

We found only one study that investigated the role of diuretics in PD emergence and no study addressing their disease-modifying effects.

#### 3.2.1. Diuretics as Neuroprotective Treatment

Warda et al. found that the at-least-once use of diuretics (although not specifically thiazides and thiazide-like diuretics) was associated with an increased risk of PD (OR = 1.23; 95% CI, 1.15–1.32) compared to never use of diuretics, but the effect was not maintained for longer therapy duration (at least three therapy years: OR = 1.12; 95% CI, 1.02–1.22; effect per therapy year: OR = 1.01; 95% CI, 1.00–1.02) [48]. 

#### 3.2.2. Diuretics as Disease-Modifying Treatment

We found no studies addressing the role of thiazide and thiazide-like diuretics in PD progression. However, there is a case report of a patient with PD who presented marked worsening of motor symptoms following exposure to spironolactone for congestive heart failure, which improved back to baseline after the withdrawal of spironolactone [54]. Damier et al. reported motor improvement (including non-dopaminergic symptoms such as freezing of gait or balance impairment) in four patients with advanced PD following use of loop diuretic bumetanide as add-on treatment to dopaminergic drugs [55]. However, these reports imply a rather symptomatic effect of diuretics (other than thiazides and thiazide-like diuretics) than an actual disease-modifying effect, and the lack of a control group hinders any association between diuretic treatment and PD progression. 

### 3.3. ACEIs

Five studies addressed the effect of ACEIs on PD emergence, whereas two studies investigated their effect on PD progression.

#### 3.3.1. ACEIs as Neuroprotective Treatment

In the study described above, Warda et al. found no association between ACEIs use and PD incidence in any of the models (OR = 1.04; 95% CI, 0.97–1.11 in model I, OR = 0.98; 95% CI, 0.91–1.07 in model II and OR = 1.00; 95% CI, 0.99–1.01 in model III) [48]. 

In a nationwide cohort study, Lee at al. followed 65,001 antihypertensive patients with different antihypertensive drugs for an average duration of 4.6 years; 8153 subjects were taking monotherapy with ACEI, with a crude incidence rate for PD of 7.81 per 1,000,000 person-days [56]. No association was found between risk of PD and ACEIs use as compared to BBs (adjusted HR = 0.80; 95% CI, 0.64–1.00) [56]. However, a decreased association was noticed with higher cumulative dosages of ACEIs (adjusted HR = 0.52; 95% CI, 0.34–0.80) [56].

Among 3637 subjects with PD, Becker et al. identified 629 patients (17.3%) who had used an ACEI prior to the index date; the relative risk estimate for current use of ACEIs compared to non-use and adjusted for body mass index, smoking status, comorbidities, diuretics, and statins was close to 1 (adjusted OR = 1; 95% CI, 0.84–1.19), whereas adjusted OR for past use of ACEIs was 0.89 (95% CI; 0.70–1.13), as compared to non-use [47]. Moreover, the ORs for users of each ACEI as well as hydrophilic (enalapril, ramipril, quinapril) or lipophilic ACEIs (captopril, lisinopril, perindopril, trandolapril, cilazapril, fosinopril) were also close to one, suggesting no association between PD emergence and ACEIs use [47]. 

In a population-based prospective study, Louis et al. considered ACEIs only for a cross-sectional analysis at baseline, where ACEIs usage did not differ between PD patients and controls (12.3% vs. 12.7%) [57]. Among patients followed prospectively (who did not have the diagnosis of PD at baseline, *n* = 30), they did not assess the risk of incident PD in ACEIs use [57]. 

In a population-based case control study, Ritz et al. noticed the direction of association between ACEIs use and PD risk to be different among ACEIs subclasses: the lipophilic ACEIs conferred a decreased risk of PD (OR = 0.90; 95% CI, 0.68–1.20), whereas the hydrophilic ACEIs increased the risk of PD (OR = 1.16; 95% CI, 0.96–1.41) [50].

#### 3.3.2. ACEIs as Disease-Modifying Treatment

In a cross-sectional study enrolling all PD patients consecutively admitted to a Day Hospital (*n* = 194), Laudisio et al. found that falls within the last 12 months were recorded less frequently in the group receiving ACEIs (OR = 0.42; 95% CI, 0.20–0.87) [53]. Even after adjusting for age and sex, the results remained statistically significant, independent of 24-h mean blood pressure levels, total dopa-equivalent dosage, and weight-adjusted daily levodopa [53]. Another interesting finding was that PD patients who took ACEIs had lower dosages of levodopa and dopaminergic agonists compared to other PD patients who had similar UPDRS, duration of disease and disability [53], suggesting a lower need for dopaminergic stimulation in this group. 

Reardon et al. evaluated the effect of perindopril on motor fluctuations in seven subjects with moderately severe PD enrolled in a double-blind placebo-controlled crossover pilot study [58]. One patient withdrew because of increased “off” periods, but the others showed significantly improved motor response to levodopa (i.e., faster onset of action, reduction in the peak-dose dyskinesia, greater amplitude of motor response) after four weeks of treatment with 4 mg perindopril [58]. This effect could not be attributed to a change in levodopa dose or pharmacokinetics [58]. 

### 3.4. ARBs

Four studies addressed the effect of ARBs on PD emergence, whereas one study investigated their effect on PD progression.

#### 3.4.1. ARBs as Neuroprotective Treatment

Becker et al. found no association between the intake of ARBs and the risk of PD (current use: adjusted OR = 1.05; 95% CI, 0.71–1.54; past use: adjusted OR = 0.76; 95% CI, 0.40–1.46) [47].

In the study conducted by Lee et al., ARBs were not found to be associated with PD risk (adjusted HR = 0.86; 95% CI, 0.69–1.08) but for higher cumulative use (adjusted HR = 0.52; 95% CI, 0.33–0.80), likewise ACEIs [56].

Warda et al. found no association between ARBs use and PD emergence in any of the models designed (at-least once used vs. never used: OR = 1.01; 95% CI, 0.93–1.09, at least three therapy years vs. less than three therapy years: OR = 0.99; 95% CI, 0.88–1.11, effect per therapy year: OR = 1; 95% CI, 0.98–1.02) [48].

Ritz et al. also found no association between ARBs use and PD risk (OR = 0.94; 95% CI, 0.74–1.19) [50].

#### 3.4.2. ARBs as Disease-Modifying Treatment

Laudisio et al. found no association between the use of ARBs and the probability/number of falls among PD patients during the last year [53]. 

### 3.5. CCBs (Dihydropyridines)

Nine studies addressed the effect of CCBs on PD emergence, whereas four studies investigated their effect on PD progression.

#### 3.5.1. CCBs as Neuroprotective Treatment

Tseng et al. performed a population-based retrospective cohort study enrolling 107,207 patients with newly diagnosed hypertension whom they followed for a median of 8.3 years. They observed that 1.2% of those treated with CCBs developed PD, as compared to 2.4% PD cases in those not exposed to CCBs [59]. Their conclusion was that the use of CCBs reduces the risk of PD (HR = 0.50) in a dose-dependent manner (HRs ranging from 0.61 to 0.37 for a cumulative defined daily dose of 90–180 to >720) [59]. 

In the study presented above, Warda et al. found no association between CCBs use and PD incidence in any of the models (OR = 0.98; 95% CI, 0.91–1.05 in model I, OR = 0.90; 95% CI, 0.83–0.98 in model II, OR = 0.99; 95% CI, 0.98–1.00) [48].

In a historical cohort study including 2,573,281 subjects, Pasternak et al. identified 202,386 users of CCBs, with a mean duration of use of 2.3 years [60]. During a mean of 7.1 person-years of follow-up, 5968 incident PD cases were detected [60]. In contrast to previous use of CCBs, current intake of CCBs was associated with a 29% more reduced risk of developing PD, after adjusting for age, sex, propensity score, and use of other antihypertensive drugs and statins (RR = 0.71; 95% CI, 0.60–0.82) [60]. Interestingly, this protective effect was more prominent among older patients [60].

In a population-based case control study, Ritz et al. also reported a 26–30% decrease in PD risk in patients prescribed centrally acting L-type CCBs, as opposed to amlodipine (acting peripherally) which did not modify the risk of developing PD [50]. They employed a two-year lag for the index date (i.e., a subject was considered exposed to medication if having received ≥2 prescriptions two years prior to the motor onset of PD) [50].

Simon et al. also investigated whether the use of CCBs was associated with the risk of PD emergence in two large prospective cohorts: 120,530 female participants in the Nurses’ Health Studies and 50,825 male participants in the Health Professionals Follow-up Study [61]. They identified 514 incident cases of PD during follow-up, but the risk of PD was not increased among subjects who reported CCBs use at baseline or during follow-up, even after adjusting for other potential PD risk factors such as arterial hypertension, physical activity, caffeine, alcohol, and total energy intake [61]. 

In the study presented above, Ton et al. found no association between CCBs use and PD risk, either for ever use or duration of treatment, dose, number of dispensed prescriptions, or pattern of use [51].

Becker et al. found that current long-term exposure to CCBs was associated with a significantly reduced risk of developing PD [47]. This effect was present in both models employed by the authors: in the first one they adjusted for use of other antihypertensive drugs, whereas in the second one they compared the use of CCBs to non-use of any antihypertensive drugs [47].

Among 65,001 hypertensive patients with a mean follow-up period of 4.6 years, Lee et al. proved the use of dihydropyridines to be associated with a reduced risk of PD compared to BBs intake (adjusted HR = 0.71; 95% CI, 0.57–0.90), especially central-acting CCBs rather than peripheral-acting ones (adjusted HR = 0.69; 95% CI, 0.55–0.87) [56]. Higher cumulative doses of felodipine and amlodipine further decreased the risk of developing PD, suggesting a potential dose-response effect of CCBs [56].

In the study mentioned above, Louis et al. included in the prospective analysis 3942 participants, out of which 30 had incident PD [57]. Baseline use of CCBs was not associated with reduced risk of incident PD [57]. The cross-sectional analysis included 5278 participants and the odds of prevalent PD with CCBs use was not statistically significant after adjusting for age, gender, education, and depressive symptoms [57]. 

#### 3.5.2. CCBs as Disease-Modifying Treatment 

Apart from the effect on PD emergence (see above), Pasternak et al. also explored the effect of CCBs on PD progression [60]. They found a significantly reduced risk of death among the PD patients using CCBs, primarily non cardiovascular somatic death, and no influence on the risk of dementia in these patients [60].

The Parkinson Study Group STEADY-PD III Investigators performed a multicenter, randomized, parallel-group, double-blind, placebo-controlled trial enrolling patients with early-stage PD (<3 years duration, without dopaminergic medication) who received 5 mg of immediate-release isradipine twice daily (*n* = 170) or placebo (*n* = 166) for 36 months [62]. Isradipine did not slow the clinical progression of PD, as reflected by changes in UPDRS at 36 months, time to onset of motor complications, quality-of-life measures, time to initiation, and use of dopaminergic medication [62]. 

Laudisio et al. found that CCBs intake was not associated with a reduced probability or number of falls (*p* = 0.572) over the last year among patients with PD, denying a potential disease-modifying or symptomatic effect of this drug class [53]. 

Marras et al. undertook a retrospective cohort study that investigates the link between dihydropyridines use and PD progression [63]. Among 4733 patients with PD and arterial hypertension, longer treatment with any dihydropyridine (more or less brain-penetrant) was associated with slower PD progression, as measured by three outcomes: time to initiation of antiparkinsonian drugs, time to application to a long-term care (a marker of functional decline), and time to death [63].

## 4. Discussion

BBs. There are conflicting results regarding the role of BBs in the emergence of PD. Two studies reported an increased risk of PD among BBs users (out of which one found the risk to be increased only in non-selective β-blockers), both taking into consideration confounders previously reported to be associated with PD risk [44,46]. The risk seemed to be dose- and time-dependent, suggesting a cumulative toxic effect paralleling the accumulation of Lewy bodies [44]. Four other studies found an association between short-term use of beta-antagonists and PD emergence, indicating a reverse causation or protopathic bias (i.e., association likely driven by the use of BBs for symptoms of early and yet undiagnosed PD such as tremor) rather than a real causative link between BBs use and PD occurrence [43,47]. Three studies addressing the same issue indicate no association between BBs use and PD emergence [45,48]. Out of these, one study reports an association between PD occurrence and propranolol exposure 1-2 years before the index date, likely due to a protopathic bias [49]. Interestingly, another study finds no link between propranolol use and PD emergence (after adjusting for possible confounders) but reports a lower risk of PD in metoprolol and carvedilol users [45]. Only one study inquires the role of BBs as disease-modifying treatment and suggests no association in this regard [53].

These conflicting findings might have to do with doses, treatment duration, comorbidities, and a myriad of molecular mechanisms. Nevertheless, current fundamental research seems to imply that BBs play a role in the pathogenesis of PD. 

Treatment with BBs (propranolol) or silencing of adrenoreceptor β_2_ gene (ADRB2) is reportedly associated with an increase in α-synuclein concentrations [5]. The proposed mechanism is via histone 3 lysine 27 acetylation, regulating promoters and enhancers of α-synuclein gene (SNCA) transcription, which results in increased α-synuclein expression, attendant aggregation, mitochondrial oxidative stress, and dopaminergic neurodegeneration [5,7]. Of note, SNCA over transcription even by small degrees seems to be employed not only in the pathogenesis of monogenic parkinsonism, but also in sporadic PD [7,64]. ADRB2 serves both as a SNCA transcriptional regulator and an encoder of the β_2_ adrenergic receptor (a member of the G protein-coupled receptor superfamily). Interestingly, this receptor works concordantly with the class C L-type calcium channel Ca(V)1.2 for full effect [65].

Apart from the α-synuclein overproduction, other mechanisms might also be involved in the interplay between the β-adrenergic system and PD. The regulation of tissue inhibitor of metalloproteinase (TIMP) and matrix metalloproteinase (MMP) levels by BBs could be of interest given the MMP overactivity in neuroinflammation with subsequent progressive dopaminergic neurodegeneration [8]. Beta-adrenergic antagonists might play an even larger role in modulating neuroinflammation in PD via the suppression of cyclooxygenase 2 (COX2), which is reportedly increased in the substantia nigra of PD patients [8]. They also seem to suppress the mitogen-activated protein kinase (MAPK) pathway, which is concerned with facilitating extracellular-intracellular communication (from signaling molecules to changes in gene expression patterns) via the extracellular signal-regulated kinase (ERK) cascade, among others. This cascade has been noted to be active in PD patients, with phosphorylated-ERK1/2 granules having been described in degenerating substantia nigra neurons [8]. Other potential mechanisms of BBs in PD pathogenesis include oxidative stress modulation and inhibition of nitric oxide synthase (NOS) expression [8].

Diuretics. The only study that addressed the effect of diuretics use in PD emergence found a positive association that was not time-dependent [48]. One explanation might relate to the PD-like symptoms exerted by diuretics—hypomagnesemia might lead to tremor, whereas hyponatremia and hypochloremia could elicit dyskinesia [48]. However, as the authors themselves acknowledge, it is not excluded that various subclasses of diuretics (such as thiazides and thiazide-like diuretics) should elicit a dose and time-dependent effect on PD emergence, but this needs further testing. Since thiazides and thiazide-like diuretics have been shown to raise the serum urate levels (which has antioxidant properties) [66], they are expected to have a neuroprotective effect on PD. We found no studies addressing the role of thiazide and thiazide-like diuretics in PD progression.

Although the anti-aggregation potency of thiazides and thiazide-like diuretics has been postulated, they have not been employed in vitro PD models to our knowledge. However, there are some interesting data related to other diuretics that emerge from experimental research and are worth mentioning. In addition to its BBB disrupting properties (by osmotic shrinkage of endothelial cells with subsequent mechanical separation of the tight junctions), mannitol interferes with α-synuclein aggregation [13]. It acts as a chemical chaperone on α-synuclein folding in vitro—high concentrations significantly decrease the tetramers and high molecular weight oligomers, whereas low concentrations affect the secondary structure of oligomers and inhibit the formation of fibrils [13]. In vivo studies reveal that mannitol reduces the α-synuclein aggregates in both PD Drosophila fly and mThy1-human α-synuclein transgenic mouse models [13]. Conversely, the pyridinium of furosemide, a metabolite found in the urine of patients treated with this loop diuretic, induces α-synuclein accumulation, reactive oxygen species, and apoptosis in human neuroblastoma cells SH-SY5Y [12]. Mice exposed to pyridinium of furosemide for seven days in drinking water revealed serine 129 phosphorylated α-synuclein, tyrosine hydroxylase decrease in striatum and tau accumulation in hippocampus [12]. The induction of neurodegeneration by this metabolite of furosemide is mediated by a specific inhibition of striatal mitochondrial respiratory chain complex I [12]. As opposed to this, the loop diuretic bumetanide has been shown to counterbalance the negative effects of dopamine deprivation by restoring the GABAergic inhibition [67]. 

ACEIs. All in all, the studies inquiring whether ACEIs use is related to PD emergence suggest that there is no association in this regard. However, one study does report a smaller risk of PD with higher cumulative dosages of ACEIs, which implies a possible dose-dependent neuroprotective effect of ACEIs [57], but this needs further confirmation. The finding that there is a difference in the risk estimates between lipophilic and hydrophilic ACEIs also warrants further investigation [50]. Both studies addressing ACEIs as potential disease-modifying treatment in PD seem to imply that ACEIs slow the progression of PD, but this conclusion is questionable since one of the studies has a cross-sectional design (which does not allow to establish a causal relationship between the use of ACEIs and fewer falls in PD) [53] and the other one only included six patients [58]. Another issue is the missing data on ACEIs dosage and the subjective assessment of falls (retrospective self-report) [53,58].

It has been hypothesized that the dysregulation of brain RAAS might be involved in neurodegeneration—elevated angiotensin II levels activate AT1 receptors, promoting neuroinflammation, oxidative stress, alterations in mitochondrial functions, glutamate excitotoxicity and reduction of cerebral blood flow with consecutive hypoxia and glucose deprivation [32,34]. The cellular damaging effects of AT1 receptors activation is mediated by the mitogen-activated protein kinase (MAPK) and c-Jun N-terminal kinase (JNK) pathways [32]. In mesencephalic cell cultures, angiotensin II activates the microglial RhoA/ROCK pathway—which upregulates microglia tumor necrotic factor, and nicotinamide adenine dinucleotide phosphate oxidase complex (NOX)—which stimulates superoxide generation, subsequently facilitating dopaminergic neurons degeneration [32,34]. The level of NOX activation is a major regulator of the shift between M1/proinflammatory and M2/immunoregulatory microglial phenotypes [36]. Apart from central angiotensin II, it is likely that the microglial polarization towards the proinflammatory phenotype [36] should also be elicited by neurons altered by α-synuclein accumulation or α-synuclein released independently of cell death [68]. Moreover, there might be a “positive feedback self-perpetuating progression of neurodegeneration” in which microglia-mediated neuroinflammation and α-synuclein-induced neuronal damage stimulate each other [68]. 

The blockade of the angiotensin II/AT1R axis by either ACEIs or ARBs has been reported to attenuate the death of dopaminergic neurons induced by the α-synuclein [32]. In PD patients, an increase in the central angiotensin converting enzyme has been noted [24,33]. In rodents’ brain, perindopril treatment increased dopamine levels in the striatal region, whereas captopril ameliorated dopaminergic degeneration via rise in 6-hidroxy dopamine (6-OHDA) levels [24]. Independently from their role in reducing angiotensin II levels, ACEIs seem to be capable of scavenging reactive oxygen species, at least in vitro [33]. Moreover, ACEIs could directly interact with the dopaminergic system, likely increasing striatal dopamine content [33].

ARBs. The conclusion of the studies that explored the effect of ARBs intake on PD emergence is that ARBs do not seem to have a neuroprotective effect in PD, but a dose-dependent effect might be involved according to Lee et al. [56]. The highest cumulative doses of ARBs were associated with a lower risk of PD, even after stratifying the population according to age and gender [56]. Although there is no report of ARBs influencing the progression of PD, it is an issue worth studying. 

Although both ACEIs and ARBs inhibit brain RAAS, their effects are superimposable only to a certain point. Animal studies have shown that ARBs enable endogenous angiotensin II to activate AT2 receptors, which stimulate neuronal regeneration and regulate pro- and/or antiapoptotic events [35]. The drugs pertaining to the class of sartans differ pharmacologically in terms of affinity for the AT1 receptor and duration of receptor blockade [35]. Generally, they have reversible action on the AT1 receptor, but candesartan dissociated extremely slow from the receptor, exerting long lasting inhibition. Although all ARBs penetrate the BBB in a dose- and time-dependent manner to a certain degree (subsequently antagonizing the brain AT1 receptors), telmisartan is more lipophilic than losartan and irbesartan and has greater potency on brain AT1 receptors following systemic administration [35]. These differences imply that the clinical studies which explore the effect of sartans on PD emergence and/or progression should address the drugs separately, taking into account the time of exposure and dosage. 

Nevertheless, whereas the exact role of nigrostriatal RAAS in the pathogenesis of PD remains to be elucidated, it is a promising target for both symptomatic and/or neuroprotective/disease-modifying effect in PD, but further clinical studies are warranted to attest this [69].

CCBs. The results of the studies that investigated the effect of exposure to CCBs on PD occurrence are equivocal. Five studies concluded that these drugs reduce the risk of incident PD, whereas four studies deny any effect on PD emergence. Moreover, the authors of one study pertaining to the former category admit that the results could reflect a symptomatic effect (by preventing the development of clinical symptoms of early disease) rather than a neuroprotective one, since they notice a rapid disappearance of effects upon discontinuation of CCBs [60]. A notable previous work on this subject is the meta-analysis by Lang et al. which includes five of the studies mentioned earlier and concludes that dihydropyridines use reduces the risk of PD by 27% (RR = 0.73; 95% CI, 0.64–0.83) [70]. However, with the studies published later, this conclusion is highly debatable. Two studies imply that CCBs intake does not alter the progression of PD. Although Marras et al. found a significant association between dihydropyridines use and PD progression, they acknowledge that this should not be considered since there is no difference between brain-penetrant dihydropyridines (i.e., felodipine and nifedipine) and those which do not substantially cross the BBB (i.e., amlodipine) [63]. Moreover, they suggest that longer treatment with CCBs impacts a broad range of outcomes (such as time to first gastrointestinal bleeding or time to first treatment with eye drops for glaucoma) which are more likely related to specific patterns of health and health care [63]. However, these results are contradicted by those of Pasternak et al., as described earlier [60], making it difficult to draw any conclusion.

Mechanisms that are either pro- or anti-parkinsonian are far from having been fully elucidated but hypotheses abound, building on the knowledge that L-type Ca 1.3(v) calcium channels that are present in substantia nigra neurons function as tyrosine hydroxylase and dopa decarboxylase enzyme-regulators and as supporters of pace making activity by the substantia nigra through calcium cytosolic oscillations [24,71]. These channels help induce sufficient ATP production to maintain the relevant pace making activity, at the price of increased mitochondrial reactive oxygen species and reactive nitrogen species [71,72,73]. With aging, these channels are reportedly upregulated, thus augmenting the burden of oxidative stress in the substantia nigra [24]. It has also been suggested that defects in calcium regulatory proteins such as calbindin and calpain, which control intracellular calcium homeostasis, might be implicated in parkinsonism [25]. Last but not least, it seems that terminus choice is also important for effective inhibition, with C-terminus mediated inhibition aiding Ca^2+^ current inactivation at its peak rather than at later stages of activation [74].

Thus, on the pro-parkinsonian side, it might be that CCBs trigger an acute decline in dopamine release, while on the anti-parkinsonian side, CCBs might help cut down on oxidative stress depending on age, channel-expression, and other interactions (regulatory proteins, relevant terminuses, etc.).

Whether any one or combinations of these purported mechanisms might act as preventive or disease-modifying avenues is equally unclear. Some research goes on to point to CCBs acting as disease modifiers based, at least in part, on the exquisite sensitivity of Cav1 channels to dihydropyridines, as well as on the dispensability of these channels in pace making maintenance—i.e., when calcium channels are blocked by dihydropyridines, pace making activity is taken over by voltage-dependent sodium, potassium, and hyperpolarization and cyclic nucleotide-activated cation (HCN) channels [26]. Yet, other papers review and stress the risk-reduction properties of CCBs in relation to PD [27,28,70,75].

Whether the molecular mechanisms of CCBs veer more towards beneficial autophagy protecting against proteinaceous aggregates or towards oxidative stress reduction, or both, is yet to be determined. It might be that no single therapeutic target holds the key to prevention and/or disease-modification in PD, as pathogenesis might be driven by an intricate interplay among calcium, cytosolic dopamine, and alpha-synuclein [29].

## 5. Limitations

A major limitation of this review relates to the impossibility of performing a quantitative analysis due to heterogeneous data found in the studies. 

Most of the studies did not report the duration of exposure to antihypertensives prior to developing PD, nor did they investigate the cumulative dosage of antihypertensives. Since the neuroprotective effect of drugs is supposedly dose- and time-dependent, this is an important drawback. A rigorous selection of patients would have caused less results compatible with reverse causation/protopathic bias. Another issue is the fact that most of the studies did not adjust for possible confounders (i.e., known risk or neuroprotective factors in PD). Some studies employed questionnaires or self-reports of previous intake of antihypertensive drugs, which could act as a bias. 

The few eligible studies that addressed the disease-modifying properties of antihypertensives in PD also have significant methodological problems: they did not perform the clinical scales (including UPDRS scale) in an “OFF” state in order to exclude the symptomatic effect of antiparkinsonian drugs and assess the real disability of the patients. Moreover, a more objective measure of disease progression such as changes in brain deposition of α-synuclein pathology would have been even more suitable.

The fact that most of the studies did not perform a separate analysis for the BBB penetrant drugs acts as bias. Although theoretically a neuroprotective effect could be exerted in the periphery (providing that the α-synuclein pathology starts in the gut and spreads to the brain by trans-neuronal propagation [76]), a disease-modifying effect most likely involves crossing the BBB. Other data that could have been relevant were not considered in any study: history of arterial hypertension, its grading, risk stratification and duration, hypertension-mediated organ damage (including vascular lesions of the basal ganglia), and emergence of orthostatic hypotension that could have resulted in the discontinuation of antihypertensive therapy. 

Noteworthy, 11 years prior, Reek et al. performed a similar systematic review [77]. Although they addressed the same issues under different names (they referred to the neuroprotective effect as primary prevention and the disease-modifying effect as secondary prevention), their methodology differs from ours: they did not include diuretics [77]. Apart from taking into consideration all the studies they had included (even though not all found by our search strategy), our systematic review adds 13 more studies into the question. Furthermore, we offer a more detailed view on the molecular mechanisms involved in the etiopathogenesis of PD that might be related to the use of antihypertensive drugs. Nevertheless, apart from the conflicting results discussed above, the assumption of our systematic review remains elusive considering that most of the studies included were observational (with a priori low quality of evidence according to the GRADE system [78]), with only two interventional studies fulfilling the inclusion criteria (out of which only one was a randomized controlled trial with high a priori quality level according to the GRADE system [78]) that enrolled a small number of patients. Since most of the observational studies included report a relative risk (RR) or hazard ratio (HR) <2 or >0.5, we can conclude that the magnitude of effect is rather small according to the GRADE system [78]. Furthermore, the cross-sectional design of some studies makes it difficult to draw any conclusions regarding the causality between antihypertensive agents’ intake and PD emergence or progression. 

## 6. Conclusions and Future Perspectives

Parkinson’s disease is a complex disorder with several pathoetiological pathways that ultimately lead to cell death. Despite substantial research efforts, there are no neuroprotective interventions or curative/disease-modifying therapies in PD. However, as James Parkinson optimistically declared, “there appears to be sufficient reason for hoping that some remedial process may ere long be discovered, by which, at least, the progress of the disease may be stopped.” [3]. With better understanding of the molecular basis of neurodegeneration in PD, we might become closer to achieving this goal. 

Considering the cascade of events involved in neuronal degeneration in PD, perhaps multi-target agents (the so-called multi-functional drugs or network therapeutics) would be able to prevent, cease, or slow this process. These kind of molecules have already shown promising results in preclinical studies as neuroprotective and disease-modifying agents [79], but regrettably these findings have not been translated into clinical trials yet. 

Although the studies addressing the potential neuroprotective and/or disease-modifying effect of antihypertensive drugs in PD reported conflicting results mainly because of heterogeneous protocols and population, there is proof that they might offer potential therapeutic solutions, but this hypothesis needs further studying and testing. Moreover, perhaps it would be wise to evaluate the effect of combination antihypertensive therapies in PD emergence and/or progression.

## Figures and Tables

**Figure 1 biomedicines-10-00653-f001:**
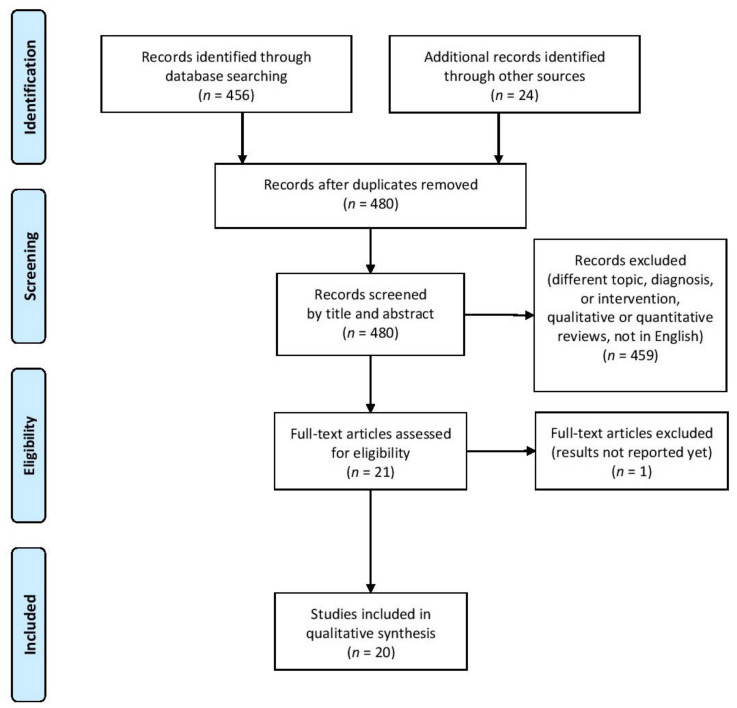
Flow diagram displaying the selection process.

**Table 1 biomedicines-10-00653-t001:** Neuroprotective effect of antihypertensive drugs in PD.

Authors	Year	Type of Study	PD Patients Exposed (No.)	Controls Exposed (No.)	Drug	Mean Follow–Up Duration (Years)	Drug Dosage (mg)	Possible Effect	Conclusion
Tseng Y.F. et al.	2021	Retrospective cohort study	832	66,588	CCBs	7.18	Cumulative defined daily dose: 0–90; 90–180; 180–360; 360–720; > 720	Yes	Treatment with CCBs was associated with a significantly reduced risk of PD in patients with newly diagnosed hypertension.
de Germay S. et al.	2020	Nested case–control study	595	561	BBs (separate analysis for propranolol)	1–2	–	No, except for propranolol	Exposure to BBs did not increase the risk of PD occurrence, except for propranolol.
Giorgianni F. et al.	2020	Cohort study with nested case–control analysis	1818	13,488	BBs	<1; 1–5; >5	–	Yes	Use of BBs was associated with an increased risk of PD, that was highest with short duration of use and decreased thereafter.
Warda A. et al.	2019	Case–control study	Model I: 53.8% –BBs; 44.8%–Ds; 43.7%–ACEIs; 18.8%–ARBs; 33.9%–CCBs	Model I: 50.9%–BBs; 38.4%–Ds; 41.1–ACEIs; 18.0%–ARBs; 32.4%–CCBs	BBs; Ds; ACEIs; ARBs; CCBs	>/= 3 vs. </= 3	–	No	No association was found between antihypertensive therapy and PD incidence.
Koren G. et al.	2019	Prospective cohort study	–	–	BBs	0.98; 1.64; 2.73; 4.1	Mean defined daily dose: 1.43	Yes	Chronic use of BBs conferred a time and dose–dependent increased risk of PD.
Hopfner F. et al.	2019	Retrospective case–control study	407	1488	BBs	0–1; 1–3; 3–5; 5–8; >/= 8	–	No	Use of BBs was associated with an increased PD risk, which was markedly stronger for short–term use compared to long–term use.
Nielsen S.S. et al.	2018	Case–control study	4.4%–propranolol; 6.9%–carvedilol; 26.3%–metoprolol	1.3%–propranolol;6.1%–carvedilol; 22.6%–metoprolol	BBs	1; 1.5	< 40; 40–80; >/= 80/day–propranolol;< 12.5; 12.5–25; >/= 25/day–carvedilol; < 50; 50; > 50/day–metoprolol	Yes	Carvedilol and metoprolol appeared to reduce PD risk.
Gronich N. et al.	2018	Nested case–control study	3032–selective BBs; 1073–non–selective BBs	27,290–selective BBs; 50,306–non–selective BBs	BBs	2; 5; 8	–	Yes	The use of non–selective BBs was associated with an increased risk of PD.
Lee Y.C. et al.	2014	Retrospective study	–	–	CCBs; ACEIs; ARBs	4.6	Any use; low dose; high dose	Yes	Centrally acting CCBs use and high cumulative doses of ACEIs and ARBs were associated with a decreased incidence of PD in hypertensive patients.
Pasternak B. et al.	2012	Historical cohort study	173	5538	CCBs	461,984 person–years	Standard dose; high dose	Yes	CCBs use was associated with a reduced risk of PD.
Simon C.K. et al.	2010	Prospective cohort study	18	52	CCBs	<4; >/= 4	–	No	No association was observed between PD risk and baseline use, frequency, or duration of CCBs use.
Ritz B. et al.	2010	Retrospective population–based case–control study	55	368	CCBs	2	–	Yes	Centrally acting CCBs use was associated with a 25–30% decrease in PD risk.
Louis E.D. et al.	2009	Case–control and prospective cohort analysis	3–BBs; 10–ACEIs; 11–CCBs; 16–Ds	187–BBs; 592–ACEIs; 514–CCBs; 861–Ds	BBs; ACEIs; CCBs; Ds	3	–	No	Antihypertensive medication use was not associated with prevalent or incident PD.
Becker C. et al.	2008	Retrospective case–control study	1704: 1168–BBs; 629–ACEIs; 89–ARBs; 807–CCBs	991–BBs; 639–ACEIs; 98–ARBs; 863–CCBs	ACEIs, ARBs, BBs, CCBs	1–9; 10–19; 20–29; 30–39; > 40 prescriptions	–	Yes (CCBs)	Current long–term use of CCBs was associated with a significantly reduced risk of PD emergence, as opposed to the intake of other antihypertensive drug classes.
Ton T.G.N. et al.	2007	Population based, case–control study	191–CCBs; 165–BBs	365–CCBs; 321–BBs	CCBs, BBs	Cumulative duration of use: CCBs: no use; < 2.5; >/= 2.5; BBs: no use; < 3; >/= 3	Cumulative standard doses: CCBs: no use; < 886; >/= 886; BBs: no use; < 4300; >/= 4300	No	No association was found between PD risk and use of CCBs or BBs in terms of duration, dose, number of prescriptions or pattern of use.

**Table 2 biomedicines-10-00653-t002:** Disease-modifying effect of antihypertensive drugs in PD.

	Year	Type of Study	PD Patients Exposed (No.)	Mean Age (Years)	Males (%)	Mean Disease Duration (mo.)	Mean H&Y Stage	Mean UPDRS Score	PD Controls (No.)	Drug	Exposure Duration (mo.)	Drug Dosage	Possible Benefit	Full Conclusion
Parkinson Study Group STEADY-PD III Investigators	2020	Multi-center, RCT	170	62.1	71.8	9.90	1.70	23.70	166	CCBs (isradipine)	36	5 mg × 2/day	No	Long-term treatment with immediate-release isradipine did not slow the clinical progression of early-stage PD.
Laudisio A. et al.	2017	Single-center, retrospective cross-sectional study	42–ACEIs; 46–ARBs; 41–BBs; 33–CCBs	73	63.9	45.3	-	43.34	152–ACEIs; 148–ARBs; 153–BBs; 161–CCBs	ACEIs; ARBs; BBs; CCBs	-	-	Yes–ACEIs; No–ARBs; No–BBs; No–CCBs	Use of ACEIs was associated with reduced probability of falling in PD patients. No association was found between use of ARBs and falls.
Marras C. et al.	2012	Retrospective cohort study	378	78.6	50.8	-	-	-	1087 (amlodipine)	CCBs–except amlodipine	> 9	-	No	CCBs did not have a clinically significant effect on the course of PD in the antihypertensive doses.
Pasternak B. et al.	2012	Historical cohort study	173	-	-	-	-	-	-	CCBs	-	-	Yes	Among patients with PD, CCBs use was associated with reduced risk of death but no dementia.
Reardon K.A. et al.	2000	Single-center, double blind placebo-controlled crossover pilot study	6	59.6	16.6	144	3	-	6 (themselves)	ACEIs–perindopril	1	2 mg/day –1st week; 4 mg/day –3 weeks	Yes	Perindopril improved motor complications in PD.

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
