# Peer review of "Shared Molecular Targets in Parkinson’s Disease and Arterial Hypertension: A Systematic Review"

_biomedicines, 2022, doi:10.3390/biomedicines10030653_

Round 1
Reviewer 1 Report
In the review article “Shared Molecular Targets in Parkinson’s Disease and Arterial Hypertension: A Systematic Review” Tulbă et. al., have assessed the role of antihypertensive agents on emergence and/or progression of Parkinson’s disease. In order to do so, a bibliographic search on the PubMed database for studies that comprise treatment of PD patients with antihypertensive drugs was performed and the outcome was divided into two categories; 1) neuroprotective (antihypertensive drug has a protective role on PD emergence) and, 2) disease-modifying (antihypertensive drug has a beneficial effect on PD progression) effect. The disease-modifying effect of antihypertensive drugs in PD reported in the various studies contradicts; which could be mainly because of heterogeneous nature of protocols and population as mentioned by authors. The information summarized in this article is going to enrich the existing knowledge to the arena of research on design of combination antihypertensive therapies having promising role on PD emergence and/or progression. This review is systematically organized with relevant information under appropriate subheadings as well as highlighting its limitations; hence, it may be accepted for publication after addressing following points.
- As this review includes plenty of acronyms so a paragraph/table describing the expanded form of various acronyms should be added.
- Table-1 and table-2 are fused in the manuscript draft I downloaded. Please make sure that the tables are properly formatted in the final version of the article to be published.
Plagiarism percentage - not checked by the reviewer.
Author Response
Thank you very much for your comments. As you suggested, we added an abbreviation list (please see line 722) and formatted the tables in order to appear separated.
Reviewer 2 Report
I love the meta analysis performed in this work. I think this is important in that it may inform future directions as to approaches that are helpful and harmful for treating PD. This is a very worthy endeavor given the devastation of this disease.
Minor edits are needed.
1) Shorten the quote from Palsy in the Introduction.
2) In the Introduction cite more references justifying the use of all of the drugs proposed in Parkinsons. For most categories one reference is cited to justify the need to investigate these agents.
3) Check for typos in the conclusion section of the summary table of drug effects.
Author Response
Thank you very much for your comments. As you suggested, we addressed the following issues:
- We shortened the quote from "An Essay on the Shaking Palsy" (please see lines 37-39).
- We cited more references that justify the investigation of these drugs in PD emergence and/or progression (please see lines 66, 75, 98, 125).
- We corrected the typos in the conclusion section of the tables.